# Mapping of Land Cover with Optical Images, Supervised Algorithms, and Google Earth Engine

**DOI:** 10.3390/s22134729

**Published:** 2022-06-23

**Authors:** Fernando Pech-May, Raúl Aquino-Santos, German Rios-Toledo, Juan Pablo Francisco Posadas-Durán

**Affiliations:** 1Department of Computer Science, Instituto Tecnológico Superior de los Ríos, Balancán 86930, Tabasco, Mexico; 2Faculty of Telematics, University of Colima, 333 University Avenue, Colima 28040, Colima, Mexico; aquinor@ucol.mx; 3Tecnológico Nacional de México Campus Tuxtla Gutiérrez, Tuxtla Gutiérrez 29050, Chiapas, Mexico; german.rt@tuxtla.tecnm.mx; 4Instituto Politécnico Nacional (IPN), Mexico City 07738, Mexico; jposadasd@ipn.edu.mx

**Keywords:** remote sensing images, land use with Sentinel-2, Sentinel-2, Sentinel-2 with Google Earth Engine

## Abstract

Crops and ecosystems constantly change, and risks are derived from heavy rains, hurricanes, droughts, human activities, climate change, etc. This has caused additional damages with economic and social impacts. Natural phenomena have caused the loss of crop areas, which endangers food security, destruction of the habitat of species of flora and fauna, and flooding of populations, among others. To help in the solution, it is necessary to develop strategies that maximize agricultural production as well as reduce land wear, environmental impact, and contamination of water resources. The generation of crop and land-use maps is advantageous for identifying suitable crop areas and collecting precise information about the produce. In this work, a strategy is proposed to identify and map sorghum and corn crops as well as land use and land cover. Our approach uses Sentinel-2 satellite images, spectral indices for the phenological detection of vegetation and water bodies, and automatic learning methods: support vector machine, random forest, and classification and regression trees. The study area is a tropical agricultural area with water bodies located in southeastern Mexico. The study was carried out from 2017 to 2019, and considering the climate and growing seasons of the site, two seasons were created for each year. Land use was identified as: *water bodies*, *land in recovery*, *urban areas*, *sandy areas*, and *tropical rainforest*. The results in overall accuracy were: 0.99% for the support vector machine, 0.95% for the random forest, and 0.92% for classification and regression trees. The kappa index was: 0.99% for the support vector machine, 0.97% for the random forest, and 0.94% for classification and regression trees. The support vector machine obtained the lowest percentage of false positives and margin of error. It also acquired better results in the classification of soil types and identification of crops.

## 1. Introduction

The world bank considers that one of the leading global concerns is food security. However, in recent years different factors such as fires, floods, and droughts have been caused by climate change, putting at risk the areas dedicated to food crops. This has caused crop cycles to be modified and agricultural production to decrease [1]. Besides, the rapid increase in the world population has generated an unprecedented additional burden on agriculture, causing the degradation of farmland, water resources, and ecosystems, thus affecting food security [2]. It is estimated that by 2050, agricultural production needs to increase by 60% to ensure food and sustenance for the population [3].

Changes in land use caused by human activities influence the alteration of ecosystems [4]. Some organizations have proposed projects to improve crop yields but with environmentally sustainable agriculture, avoiding soil deterioration to address this situation [5,6].

On the one hand, because Mexico has a diversity of climates and massive extensions of farmland, agriculture is one of the country’s economic activities. Thus, Mexico produces a great variety of agricultural products. The agricultural production of Mexico covers 4% of gross domestic product (GDP) [7]. In recent years, the demand for farming foods has increased, causing overexploitation of natural resources. In Mexico, extreme droughts and severe floods have been recorded that have caused the loss of large extensions of crops, reducing their production. For this reason, it is of great importance to obtain information, map and identify crop areas that allow the development of strategies that counteract the effects of climate change on crops, develop sustainable agriculture, develop strategies that strengthen the field, and evaluate projects already implemented, in addition to estimating agricultural production. Therefore, it is necessary to obtain multitemporal data that monitor and identify crops, climate change, and human activities.

Currently, techniques and tools are being developed to monitor the Earth’s crust and determine changes in vegetation. Remote sensing is the science that collects information about the Earth’s surface, providing valuable data for land-use mapping, crop detection, etc. [8,9,10]. Artificial intelligence includes machine learning algorithms for land-use classification through satellite images [11].

The satellites that orbit the Earth provide unique information for additional research such as natural disasters, climate change, crop monitoring, etc. They use optical, infrared, and microwave sensors. Optical sensors provide high-resolution and multispectral images. Microwave sensors provide SAR (Synthetic Aperture Radar) prints with higher resolution and can operate in any weather condition. Many approaches analyze land cover using optical images. However, these images may contain noise (cloud cover). Some methods use SAR images. However, these images require more processing.

Spectral data from optical sensors is highly correlated with the Earth’s surface, and image analysis algorithms are mainly based on visual data. Therefore, they are primarily used for land-cover analysis.

With the advancement of satellite programs, the spatial, temporal, and infrared spectral resolution have improved significantly. New indices have been developed for land-cover analysis.

This paper proposes a methodology to map corn and sorghum crops by Sentinel-2 satellite imagery, reflectance index calculations, and supervised machine learning methods. The study area belongs to the state of Tabasco, Mexico. The document is structured as follows: Section 2 describes the theoretical framework and related works; Section 3 describes the materials and methods used in the research; Section 4 presents the results of the experiments; and finally, Section 5 contains the conclusions derived from the study.

## 2. Background and Related Works

### 2.1. Remote Sensing

Remote sensing (RS) is the science responsible for collecting information from an object, area, or phenomenon without direct contact with it through sensors that capture the electromagnetic radiation emitted or reflected by the target [12,13]. Earth observation satellites orbiting the planet record the electromagnetic radiation emitted by the Earth’s surface. Its operation is based on spectral signatures (the ability of objects to reflect or emit electromagnetic energy). With spectral signatures, it is possible to identify different types of crops, water bodies, soils, and other characteristics of the Earth’s crust.

RS has evolved from visible wavelength analog systems based on aerial platforms to digital systems using satellite platforms or unmanned aerial vehicles with coverage on a global scale [14]. The sensor resolution is the ability to record and separate information and depends on the combined effect of different components. The solution involves four essential characteristics: (1) spatial, which characterizes the Earth’s surface that each pixel of an image represents; (2) spectral, number, and bandwidth of the electromagnetic spectrum that can be recorded; (3) temporal, which determines the time it takes to obtain an image of the same place with the same satellite; and (4) radiometric, which represents the different digital levels used to record radiation intensity.

The use of satellite images may be limited by the type of passive sensors they use: (1) sensors that operate in the optical range and (2) microwave electromagnetic spectrum. The optical sensors provide multispectral images with 13 bands with characteristics that differentiate geological components such as water, vegetation, cloud, and ground cover. However, they can be affected by clouds or rain [15]. This makes it impossible to acquire images without cloudiness. The microwave sensors provide images that are not affected by weather conditions since they operate at longer wavelengths and are independent of solar radiation. However, spatial resolution and complicated processing technologies and tools limit their use [16].

Optimal optical images for vegetation identification, mapping, and atmospheric monitoring are derived from multispectral sensors [17]. Data for monitoring and analysis at the local level are obtained by drones or airplanes. In contrast, data from dedicated satellite platforms are used for ground monitoring [18].

Space programs dedicated to Earth observation include Landsat [19], Aqua [20], Copernicus [21], and more.
Copernicus Sentinel. It is a series of space missions developed by the European Space Agency (ESA) that observe the Earth’s surface and are composed of five satellites with different objectives [21]: (1) Sentinel-1 focuses on land and ocean monitoring; (2) Sentinel-2 has the mission of Earth monitoring; (3) Sentinel-3 is dedicated to marine monitoring; (4) Sentinel-4’s main objective is the measurement of the composition of the atmosphere in Europe and Africa; and (5) Sentinel-5 measures the atmospheric composition.

### 2.2. Sentinel-2 Project

The Sentinel-2 mission monitors the Earth’s surface with two satellites with similar characteristics (Sentinel 2A and 2B) that have an integrated 13-bands MSI (Multi-Spectral Instrument) optical sensor (see Table 1) that allows the acquisition of high-spatial-resolution images [22]. Each of the Sentinel-2 images covers a 290 km strip that, combined with its resolution of 10 to 60 meters per pixel and the 15-day review frequency on the equator, means that 1.6 Tbytes of image data are generated daily [23].

Among the Sentinel-2 mission objectives are:Provide global and systematic acquisitions of high-resolution multispectral images with a high review frequency.Provide continuity of the multispectral images provided by the SPOT satellites and the LANDSAT thematic mapping instrument of the USGS (United States Geological Survey).Provide observational data for the next generation of operational products, such as land-cover maps, land change detection maps, and geophysical variables.

Due to its characteristics, Sentinel-2 images can be used in different research fields such as water body detection [24] and land-cover classification [25]. In addition, Sentinel-2 photos can be combined with images from other space projects such as SPOT 4 and 5 that allow for historical studies.

Sentinel-2 spectral bands provide data for land-cover change detection/classification, atmospheric correction, and cloud/snow separation [26]. It is essential to mention that the MSI of Sentinel-2 supports many Earth observation studies and programs. It also reduces the time needed to build a European cloud-free image archive.

The MSI works by passively collecting reflected sunlight from Earth. The incoming light beam is split by a filter and focused onto two separate focal plane arrays inside the instrument: one for the visible and near-infrared (VNIR) bands and one for the short-wave infrared (SWIR) bands. The new data are acquired as the satellite moves along its orbital path.

### 2.3. Reflectance Indices

Reflectance indices are dimensionless variables that result from mathematical combinations involving two or more spectral bands. The reflectance indices are designed to maximize the characteristics of vegetation and water resources but reduce noise [27,28].

This allows analyzing the activity of vegetation and water bodies showing their seasonal and spatial changes. The most used indices in RS are:Normalized Difference Vegetation Index (NDVI) [29]. An indicator of photosynthetic biomass that calculates vegetation’s health is highly related in studies under drought conditions [30,31]. Its range is between +1 and −1. The highest value reflects healthy and dense vegetation; the lowest value reflects sparse or unhealthy vegetation. The NDVI is calculated using the following formula:
(1)$$NDVI=\frac{\left(NIR-RED\right)}{\left(NIR+RED\right)}$$
where NIR corresponds to the near-infrared band and RED to the red band.Green Normalized Vegetation Index (GNDVI). It is a modified version of NDVI that increases the sensitivity to variations in the chlorophyll of the vegetation [32]. It is calculated using the following formula:
(2)$$GNDVI=\frac{\left(NIR-GREEN\right)}{\left(NIR+GREEN\right)}$$
where NIR corresponds to the near-infrared band and GREEN to the green band.Enhanced Vegetation Index (EVI). It is an indicator that allows quantifying the greenness of the vegetation, increasing the sensitivity of the regions with a high presence of vegetation and correcting atmospheric conditions that cause distortions such as aerosols [33,34]. The formula to calculate it is:
(3)$$EVI=G*\frac{\left(NIR-RED\right)}{\left(NIR+C1*RED-C2*BLUE+L\right)}$$
where *L* is used for the soil adjustment factor; C1 and C2 are the coefficients used in the blue band to correct for the presence of the aerosol in the red band; *G* corresponds to the profit factor; and RED, NEAR, and BLUE correspond to the red, near infrared, and blue bands, respectively.Soil Adjusted Vegetation Index (SAVI). It is used to suppress the effect of the soil in areas where the vegetative cover is low, minimizing the error caused by the variation of the soil brightness [35]. The formula to calculate it is:
(4)$$SAVI=\frac{\left(NIR-RED\right)}{\left(NIR+RED+L\right)}*\left(1+L\right)$$
where *L* is the ground adjusted factor, NIR is the near-infrared band, and RED is the red band.Normalized Difference Water Index (NDWI). It is sensitive to changes in the content of water resources and is less susceptible to the atmospheric effects than affect NDVI, and it is widely used in the analysis of water bodies [36]. It is calculated using the following formula:
(5)$$NDWI=\frac{\left(NIR-SWIR\right)}{\left(NIR+SWIR\right)}$$
where NIR corresponds to the near-infrared band, and SWIR refers to the short wave infrared band.

### 2.4. Satellite Image Classification Algorithms

Image classification is used in many works in RS. Multiband imagery is widely used to map land use and recognize areas of crops, forests, bodies of water, etc. The use of predictive machine learning methods makes it possible to identify patterns contained in the images. Generally, a distinction is made between supervised and unsupervised classification.

Supervised classification techniques work as part of a group of elements belonging to the image, known as training areas. The classification of the image set is the process by which each piece contained in the picture is assigned a category based on the attributes in the training areas. The supervised classification forces the result to correspond to land covers defined by the user and, therefore, of interest to them. However, it does not guarantee that the classes are statistically separable [37].

Unsupervised classification methods perform an automatic search by grouping uniform values within an image. From the digital levels, it creates several clusters with pixels with similar spectral behavior. It is important to note that the analyst must indicate the thematic meaning of the generated spectral classes since the program does not detect it [37].

Due to the interest in classification, many automatic classifiers have been developed that can be used in the SR area. Some of the most used algorithms are:Maximum Likelihood. It starts from the assumption that the reflectivity values in each class follow without a multivariate normal probability distribution, which uses the vector of means and the variance–covariance matrix to estimate the probability that a given pixel belongs to each class. The pixel will finally be assigned to the class whose membership probability is higher. Once the assignment of pixels to the classes is finished, probability thresholds are established for each category, rejecting the pixels with a very low probability [38].Support Vector Machine (SVM). This method was developed from the statistical learning theory, which reduces the error related to the size of the training or sample data [39,40]. It is a machine learning algorithm used in problems where input–output vector dependencies such as image classification and linear regression are unknown [41].Random Forest (RF). It is a classification algorithm that aims to counteract variations in predictions in a decision tree caused by disturbances in training data [42]. The algorithm is designed so that the predictor trees produce as many errors as possible, thus ensuring that the rest of the classifiers reject it, improving their precision [43]. This algorithm has been widely used in remote sensing for land-cover classification [44].Classification and Regression Trees (CART). It is a nonparametric machine learning method [45]. Create a predictive tree using binary division until the rule for inductive detection of relationships between input and output attributes is met. Used in prediction and classification problems, the constructed trees are optimized to obtain the best prediction possible [46].

### 2.5. Google Earth Engine

Google Earth Engine (GEE) (https://developers.google.com/earth-engine, accessed on 1 March 2022) allows high-performance computational resources to process extensive referenced data collections [47]. GEE has a robust repository of free access geospatial data that includes data from various spatial projects such as Sentinel images [48,49], Landsat [50], and climate data [51], among others [52,53,54]. This web platform facilitates the development and execution of algorithms applied to collections of georeferenced images and other data types.

### 2.6. Related Works

Several approaches to vegetation mapping have been explored. The ones mentioned here generally use Sentinel satellite imagery and spectral indices.

Shaharum et al. [55] presented an oil palm mapping plantations by Landsat 8 satellite images. The study period was 2016 and 2017. They used NDVI and NDWI spectral indices. They used NDVI and NDWI spectral indices and three classification methods: (1) random forests, (2) classification tree and regression, and (3) support vector machines. The data and methods were processed in GEE. The results obtained demonstrated the capacity of GEE for data processing and the generation of high-precision crop maps. Furthermore, they mapped the land cover of oil palms. They got 80% overall accuracy in each of the methods used.

Borrás et al. [56] present research that addresses two objectives: (1) determine the best classification method with Sentinel-2 images; (2) quantify the improvement of Sentinel-2 concerning other space missions. They selected four automatic classifiers (LDA, RF, Decision Trees, KNN) applied in two agricultural areas (Valencia, Spain, and Buenos Aires, Argentina). Based on the Kappa Index, they obtained a land-use map from the best classifier. They determined that the best classifiers for Sentinel-2 images are KNN and the combination of KNN with RF. They obtained 96.52% overall accuracy. Detection of abandoned soils and lucerne was better.

On the other hand, Liu et al. [57] present a pixel-based algorithm and phenological analysis to generate large-scale annual crop maps in seven areas of China. They used Landsat 8 and Sentinel-2 images from 2016 to 2018. They use GEE for image processing and several spectral indices to examine the phenological characteristics of the crop. The results show the importance of spectral indices for crop phenological detection. In addition, they allowed working with different image repositories in GEE. Overall accuracy was 78%, 76%, and 93% using Landsat and Sentinel-2. Detection of abandoned soils and lucerne was better.

Ashourloo et al. [58] presented a method for mapping the potato crop in Iran in 2019. They analyzed and used Sentinel-2 images and the machine learning method SVM and Maximum Likelihood (ML). The method is based on the potato’s spectral characteristics during its life cycle. The results show that SVM obtains better results—an overall accuracy of better than 90% in the study sites. Finally, Macintyre et al. [59] tested Sentinel-2 images for use in vegetation mapping. They used SVM, RF, Decision Tree (DT), and K-Nearest Neighbors (KNN) algorithms. The algorithm that obtained the highest performance in the classification was SVM. Furthermore, the authors state that Sentinel-2 is ideal for classifying vegetation composition. The obtained results were: SVM (74%), KNN (72%), RF (65%), and CT (50%).

Hudait et al. [60] mapped the heterogeneous crop area according to the crop type in the Purba Medinipur District of West Bengal. They used Sentinel-2 multispectral imagery and two machine learning algorithms: KNN and RF. Plot-level field information was collected from different cropland types to frame the training and validation datasets for cropland classification and accuracy assessment. The maps obtained allowed us to identify the cultivated surfaces of Boro rice, vegetables, and betel. They got 95% overall accuracy. The study showed that RF is the more accurate.

Silva et al. [61] developed an algorithm (phenology-based) for soybean crop mapping by spectral indices and Landsat and Sentinel-2 images. The study season was 2016–2017. The algorithm is based on the soybean’s phenology during their growing cycle. Therefore, they divided their life cycle into two stages: (1) vegetative and (2) reproductive. The results demonstrate the difficulty of obtaining many images with little noise in the study area. On the other hand, the images acquired by the MODIS sensor (from the Terra satellite program) [32] were slightly better than MSI. However, MSI images have better resolution.

## 3. Materials and Methods

The methodology applied for mapping crops is divided into five stages (see Figure 1) described below.

### 3.1. Location

The study area is located in the eastern part of Tabasco, Mexico (see Figure 2a). Approximately between latitude 17${}^{{}^{\circ}}$15${}^{\prime }$29.7329${}^{\prime \prime }$ N, y 18${}^{{}^{\circ}}$10${}^{\prime }$45.0525${}^{\prime \prime }$ N, and between longitude 90${}^{{}^{\circ}}$59${}^{\prime }$12.4464${}^{\prime \prime }$ O y 91${}^{{}^{\circ}}$44${}^{\prime }$22.1932${}^{\prime \prime }$ O. The area includes the towns of Balancán, Emiliano Zapata, and Tenosique, with an approximate size of 6079 km${}^{2}$ (see Figure 2b). It has large volumes of aquifers and sediments collected by streams, rivers, and lagoons; the region’s climate is hot-humid with abundant rains in summer; its mean annual temperature is 26.55 ${}^{{}^{\circ}}$C; the average humidity is 80% and maximum 85%. Due to the terrain and climate, the main activities are cattle ranching and agriculture, with corn, sorghum, and sugar cane growing.

Data. Sentinel-2 satellite images with the Google Earth Engine (GEE code) platform through the Copernicus/S2 repository. Because crop coverage is identified in the different seasons of the year, time series per year were created considering the crop cycles and weather type of study area. The images were selected in two annual time series: (1) Spring–Summer (20 March–20 October) and (2) Autumn–Winter (21 October–20 March), from 2017 to 2019, obtaining six collections of images.

To delimit the study area, a shapefile file obtained from the National Commission for the Knowledge and Use of Biodiversity (CONABIO) [62] was used. An images fitering was applied with less than 20% clouds to obtain better images. Thus, 309 images were obtained (see Table 2).

### 3.2. Image Selection

To obtain cleaner and sharper images, pixels with small accumulations of clouds (dense and cirrus) were removed by cloud masking using the QA60 band. The thick clouds were identified by the reflectance threshold of the blue band, and to avoid erroneous detection (e.g., snow), the SWIR reflectance and the Band 10 reflectance were used. For identification of cirrus clouds, a filter was applied based on morphological operations in dense and cirrus masks: (1) erosion, to eliminate isolated pixels, and (2) dilation, to fill the gap and extend the clouds.

### 3.3. Preprocessing

Spectral indices were calculated for collections of masked images. Spectral indices are based on vegetation’s red and infrared spectral bands and electromagnetic energy interactions. For vegetation detection, the following were calculated: Normalized Difference Vegetation Index (NDVI), Green Normalized Difference Vegetation Index (GNDVI), Improved Vegetation Index (EVI), Soil Adjusted Vegetation Index (SAVI), and Normalized Difference Moisture Index (NDMI). For water bodies: Normalized Difference Water Index (NDWI). The Sentinel-2 bands used for each spectral index are:(6)$$NDVI=\frac{\left(B8-B4\right)}{\left(B8+B4\right)}$$
(7)$$GNDVI=\frac{\left(B8-B3\right)}{\left(B8+B3\right)}$$
(8)$$EVI=\frac{\left(2.5*(B8-B4)\right)}{\left(B8+6*B4-7.5*B2+1\right)}$$
(9)$$SAVI=\frac{\left(B08-B04\right)}{\left(B08+B04+0.428\right)}*\left(1.428\right)$$
(10)$$NDWI=\frac{\left(B3-B8\right)}{\left(B3+B8\right)}$$
(11)$$NDMI=\frac{\left(B8-B11\right)}{\left(B8+B11\right)}$$

For image correction, mosaics were formed by cutting out the contour of the study area and a reduction method by histograms, and linear regression (supplied by GEE through the ee. Reducer class) was applied to allow the data aggregation over time. This required reducing the image collection (input) to a single image (output) with the same number of bands as the input collection. Each pixel in the output image bands contains summary information for the pixels in the input collection. To provide additional information to the classification methods on the dynamic range of the study area, five percentages (10%, 30%, 50%, 70%, and 90%) and the variance of each band that composes the reduced image were calculated. The electron spectrum is recorded by placing the minimum, medium, maximum, and intermediate points to form a 78-band image.

### 3.4. Supervised Classification

In the classification stage, the study area’s main types of land were identified. This was done by visual analysis of satellite images, vegetation maps, and crop estimation maps obtained from the agricultural and fishing information service (SIAP, Ministry of Agriculture and Rural Development of Mexico).

Two types of crops (corn and sorghum) and six types of land use were identified: water masses (extensions of water), lands in recovery (grounds without sowing with little or no presence of vegetation), urban areas (towns or cities), sandy areas (accumulation of mineral or biological sediments), forests or tropical jungle (zone with a high vegetation index), and others (grasslands, etc.). For crops and soil types identification, three supervised classification algorithms were applied: Random Forest (RF), Support Vector Machines (SVM), and Classification and Regression Trees (CART). Supervised learning classification methods require datasets labeled with land-use categories for learning and training. GeoPDF (https://www.gob.mx/siap/documentos/mapa-con-la-estimacion-de-superficie-sembrada-de-cultivos-basicos, accessed on 3 January 2022) (estimation of crop sowing area) documents and Google Earth files provided by SIAP (https://datos.gob.mx/busca/dataset/estimacion-de-superficie-agricola-para-el-ciclo-primavera–verano, accessed on 24 August 2021) with hydrographic maps and vegetation maps and visual identification were selected to compose the training dataset.

Crop cycles and seasonal climate change cause differences in spectral indices in crops and soil types, leading to misclassifications. Therefore, it was decided to form independent datasets corresponding to each crop cycle. To address this issue, two separate data sets were created using sample points or pixels corresponding to each growing process. The pixels of the spring–summer and autumn–winter cycles were selected and entered manually in GEE based on the collection of images from 2019 and 2018 (see Figure 3), forming two datasets with 2510 sample points for spring–summer and 3012 for autumn–winter (see Table 3).

Considering the data-driven framework of machine learning models to evaluate the performance and accuracy of classification methods [63] and avoid overtraining, the dataset was divided into 70% for the training set and 30% to evaluate the performance and accuracy of classification methods.

The SVM, RF, and CART classification algorithms were evaluated and executed with different configurations on the GEE platform to improve classification efficiency.

For SVM, a kernel with a radial and gamma base function of 0.7 was used with a cost of 30. Two pieces of training were carried out: spring–summer and autumn–winter. RF was configured so that the random forest limits 20 trees and avoids misclassifications; this configuration obtained significant improvements. The base GEE configuration was used with CART since it acquired a lower number of classification errors.

## 4. Results Evaluation

From the data, two categories were defined: (1) types of crops and (2) types of land use. Corn (CC) and sorghum (SC) are found in crops. Soil types are water bodies (WB), land in recovery (LR), urban areas (UA), sandy areas (SA), tropical rainforest (TR), and others.

For the test of the classified maps, 30% of the sample points were used: 742 for the spring–summer season and 868 for the autumn–winter season (see Table 4).

The overall training accuracy (OA) and the kappa index (KI) were calculated for each season and classification method. Table 5 shows that SVM obtained the best performance in both seasons; OA and KI were 0.996%. The RF method brought an OA and a KI greater than 0.990 in the spring–summer season; in the autumn–winter season, it was 0.96% and 0.95%, respectively. Lastly, the CART method obtained an OA of 0.94% and a KI of 0.92% in the first season, and in the second season, it received 0.98% and 0.97%, respectively. Values closer to 1 indicate better performance, and therefore, the results are more reliable, while values relative to 0 indicate unreliable results.

### Coverage of Sorghum and Corn Crops with Government Data

The SIAP oversees collecting crop data. However, these data only consider the hectares planted. Consequently, those that do not sprout or do not grow are ignored. That makes these data unreliable. As a result, the margins of error of the hectares detected by the algorithms and the SIAP data are enormous.

The types of crops were compared with data obtained from the SIAP. Table 6 shows the hectares of produce for the spring–summer (s-s) and autumn–winter (a-w) seasons.

Figure 4 shows the maps generated by the SVM method. Table 7 shows the results of the estimation of the coverage of the crop types and land use using the SVM method. Results are reported in square hectares. They are classified by municipality (zone) and in two seasons of each year: spring–summer (s-s) and autumn–winter (a-w). The gray cells indicate the extensions with the highest coverage, corn in 2019 autumn–winter (1514.59 ha) and sorghum in 2017 autumn–winter (348.11 ha) for zone 1. For zone 2, it was corn in 2018 spring–summer (11,856.54 ha) and sorghum in 2017 autumn–winter (4248.01 ha).

Figure 5 shows the maps generated by the RF method, and Table 8 shows the results in land cover.

Finally, Figure 6 and Table 9 show the results obtained with the CART method.

The predictions of the three classifiers were compared with the ground truth provided by SIAP. Percentage errors for each classifier are shown in Table 10. The results obtained by SVM were superior to the actual data. The SVM method received a 5.86% general error in corn and 9.55% in sorghum crops. On the other hand, the accurate data may have a margin of error because some lands may be cultivated occasionally. This means that small crops or lands where crops are intermittent are not accounted for.

## 5. Discussion

We obtained that optical satellite images are beneficial for land and land-cover maps. Some approaches that use the same technologies and tools for the land-cover map are [48,50,64].

These images have characteristics that allow different research types in various fields to be carried out. However, Sentinel-2 photos are obtained through passive sensors; they usually present cumulus clouds that make it difficult to collect scenes in areas where the high frequency of cloudiness prevents the taking of large amounts of images. In the southeast of Mexico, specifically the state of Tabasco, as it has a high humidity index, large amounts of clouds are frequent, making investigations using Sentinel-2 images difficult, which makes it necessary to be preprocessed to obtain cleaner images. On the other hand, supervised classification methods can perform soil classifications. All this is according to the configuration, and data sets used.

Some studies in the literature used Sentinel-2 and the three classification algorithms mentioned. Praticó et al. [48] and Loukika et al. [64] used Sentinel-2 and the RF, SVM, and CART algorithms. The best results obtained were with RF and SVM. Both our approach and that of Praticó et al. and Loukika et al. used NDWI for water body detection and vegetation NDVI, GNDVI, and SAVI. The processing tool was GEE.

It is important to note that for NDVI and NDWI, training points and polygons were created for each class, and each pixel within the polygon represents training data. Since the assigned value for each pixel is known, we can compare them with the classified ones and generate an error and precision.

It should be noted that a bagging technique was applied for the RF training. For SVM, an instance was created that looks for an optical hyperplane separating the decision boundaries between different classes. RF and SVM receive the training data, detectable types, and spectral bands (bands 2, 3, 4, 5, 8, 11, NDVI, NDWI). Furthermore, in RF, the number of trees and variables in each split is needed, while in SVM, the Gamma costs and kernel functions are required [65].

On the other hand, Tassi et al. [50] analyze land cover by Landsat 8 images, RF, and GEE. They use two approaches: pixel-based (PB) and two object-based (OB). SVM and RD are the algorithms with the best results in these mentioned approaches.

The three mentioned approaches, as well as our proposal, use supervised algorithms for land-cover classification. They also use the Google Earth Engine for image processing. The results obtained from the three approaches are like our proposal. They also use the same spectral indices for land-use and land-cover maps. The evidence presented above demonstrates the importance of Sentinel-2 satellite imagery in the field of soil classification and crop detection. Sentinel-2 images have characteristics that allow different investigations to be carried out in different fields.

However, Sentinel-2 images usually present cumulus clouds that make it difficult to collect scenes in areas where cloudiness is high, preventing the taking of large amounts of photos. This is because passive sensors obtained them, making it necessary to be preprocessed to get cleaner images.

## 6. Conclusions

Sentinel-2 satellite images have characteristics that allow them to be used in land-use clasification, crop detection, and different research fields. However, since they are obtained through passive sensors, they can present cumulus clouds that make it difficult to collect scenes in gray areas. The area and seasons studied presented a high rate of humidity, which made the research difficult. On the other hand, the execution capacity of the Google Earth Engine platform proved to be effective in land-use analysis and classification. The methods used for land-use classification and crops of sorghum and corn were SVM, RF, and CART, which obtained different results. SVM obtained 0.99%, RF 0.95%, and CART 0.92% overall accuracy. SVM had the lowest percentage of false positives and the lowest margin of error compared to the real data. According to the data obtained, the corn crop has the greatest presence in the study area, and sorghum has a decreased presence.

Food production in the study area does not show significant changes. Compared to population growth, production is inefficient, which is a risk to food security in the area. This makes it necessary to import products.

Future work intends to improve the sample datasets to have a better data range, use unsupervised learning methods, and use SAR data (Sentinel-1) and other satellites to increase the images and build maps with greater precision.

## Figures and Tables

**Figure 1 sensors-22-04729-f001:**
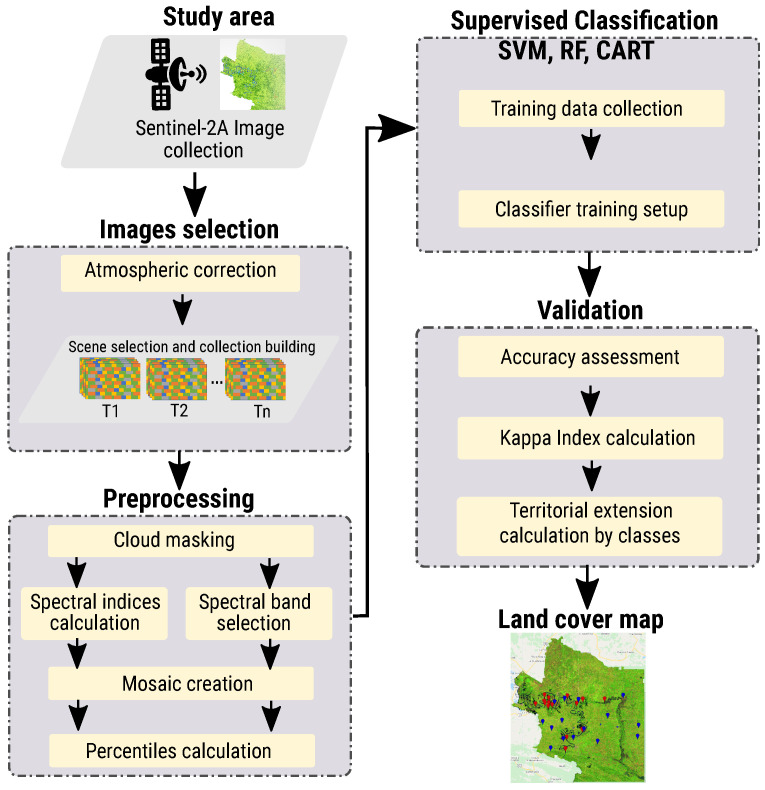
Proposed methodology for land-cover classification.

**Figure 2 sensors-22-04729-f002:**
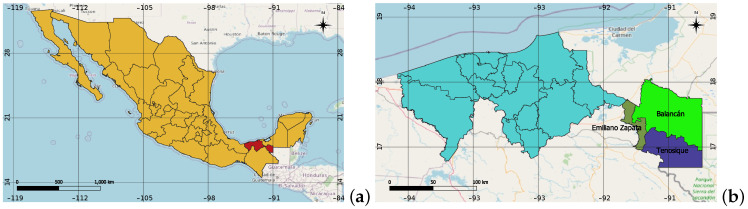
Study area map’s. (**a**) Tabasco in Mexico; (**b**) Study area.

**Figure 3 sensors-22-04729-f003:**
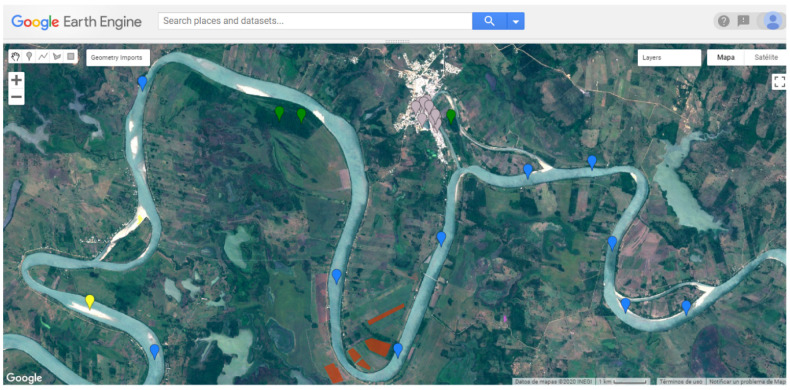
Sample points on the GEE platform.

**Figure 4 sensors-22-04729-f004:**
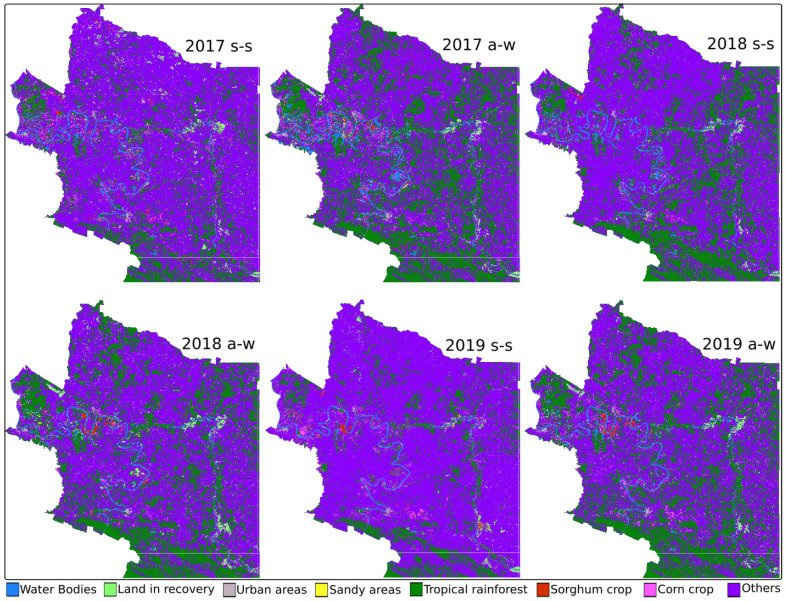
SVM-generated maps.

**Figure 5 sensors-22-04729-f005:**
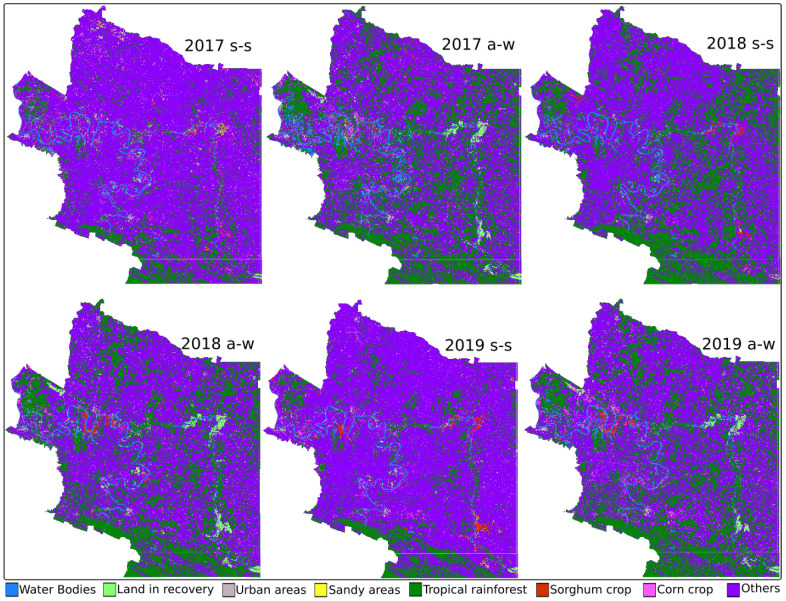
RF-generated maps.

**Figure 6 sensors-22-04729-f006:**
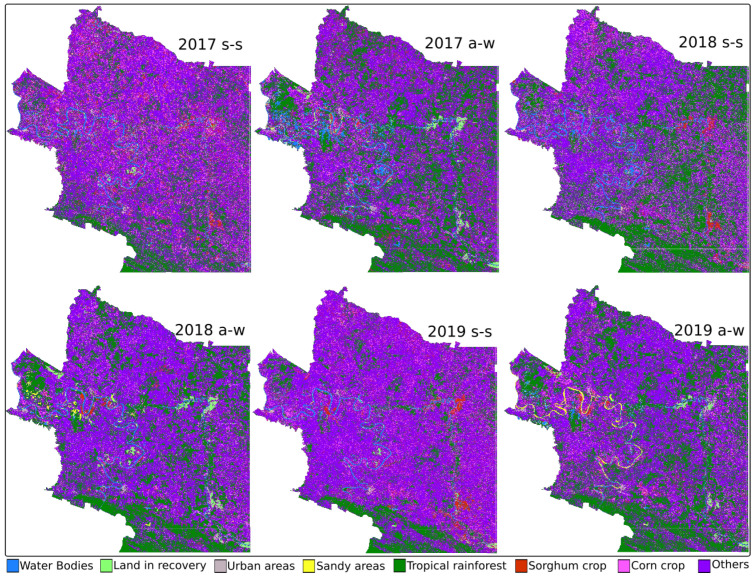
CART-generated maps.

**Table 1 sensors-22-04729-t001:** Spectral bands for Sentinel-2A and Sentinel-2B sensors. Bold bands were used in this research.

	Sentinel-2A		Sentinel-2B	
**Band**	**Wavelength (nm)**	**Resolution (m)**	**Wavelength (nm)**	**Resolution (m)**
1 Coastal aerosol	443.9	60	442.3	60
**2 Blue**	496.6	10	492.1	10
**3 Green**	560	10	559	10
**4 Red**	664.5	10	665	10
**5 Vegetation red edge (VNIR)**	703.9	20	703.8	20
6 Vegetation red edge (VNIR)	740.2	20	739.1	20
7 Vegetation red edge (VNIR)	782.5	20	779.7	20
**8 Near infrared (NIR)**	835.1	10	833	10
8a Narrow NIR	864.8	20	864	20
**9 Water vapor**	945	60	943.2	60
10 Short-wave infrared (SWIR) cirrus	1373.5	60	1376.9	60
**11 SWIR**	1613.7	20	1610.4	20
12 SWIR	2202.4	20	2185.7	20

**Table 2 sensors-22-04729-t002:** Time series imaging dataset.

Season	Img 2017	Img 2018	Img 2019
Spring–Summer	12	70	66
Autumn–Winter	46	63	52

**Table 3 sensors-22-04729-t003:** Sample points collection.

Coverage	Spring–Summer	Autumn–Winter
Corn crop	194	190
Sorghum crop	969	822
Water bodies	324	324
Land in recovery	152	628
Urban areas	100	100
Sandy areas	191	233
Tropical rainforest	353	378
Others	227	337

**Table 4 sensors-22-04729-t004:** Collection of sample points for test.

Coverage	Spring–Summer	Autumn–Winter
Corn crop	56	57
Sorghum crop	285	262
Water bodies	95	95
Land in recovery	49	177
Urban areas	28	38
Sandy areas	54	63
Tropical rainforest	112	125
Others	63	107

**Table 5 sensors-22-04729-t005:** Overall accuracy (OA) and Kappa index (KI) of the seasons.

	RF		SVM		CART	
**Season**	**OA**	**KI**	**OA**	**KI**	**OA**	**KI**
Spring–Summer	0.9671	0.9580	0.9973	0.9966	0.9426	0.9260
Autumn–Winter	0.9920	0.9904	0.9988	0.9986	0.9815	0.9777

**Table 6 sensors-22-04729-t006:** Coverage in hectares of corn and sorghum crops provided by SIAP for the spring–summer (s-s) and autumn–winter (a-w) seasons.

	Emiliano Zapata		Balancán		Tenosique	
**Year**	**Corn**	**Sorghum**	**Corn**	**Sorghum**	**Corn**	**Sorghum**
2017 s-s	1045	298	10,267	2538	2860	no data
2017 a-w	1340	323	7577	4025	2146	339
2018 s-s	1017	37	11,654	107	4262	no data
2018 a-w	1370	95	7926	3872	1738	262
2019 s-s	1175	50	11,357	1606	4305	120
2019 a-w	1425	115	8086	3972	2485	421

**Table 7 sensors-22-04729-t007:** Land-use coverage classified by SVM. Coverage in hectares.

Season	CC	SC	WB	LR	UA	SA	TR	Others
Zone 1. Emiliano Zapata								
2017 s-s	1128.31	318.12	2697.58	445.77	369.16	23.54	15,263.43	38,985.48
2017 a-w	1422.23	348.11	4441.97	1141.84	441.71	224.77	21,411.62	29,799.01
2018 a-w	1409.32	104.99	2499.00	820.43	343.68	432.84	22,769.34	30,354.33
2019 s-s	1238.68	57.12	1515.61	339.56	374.70	47.11	9651.58	44,665.69
2019 a-w	1514.59	124.28	2391.12	1079.26	420.81	92.18	21,616.85	31,992.72
Zone 2. Balancán								
2017 s-s	10,572.43	2720.76	5573.76	9431.75	1971.42	64.75	49,028.55	278,072.51
2017 a-w	7820.23	4248.01	7457.56	5339.38	648.59	183.14	118,043.15	213,995.85
2018 s-s	11,856.54	112.31	6350.09	2402.60	449.38	26.99	95,198.04	245,288.92
2018 a-w	8354.53	4060.53	5575.26	6459.76	623.32	528.99	100,175.72	231,858.86
2019 s-s	11,763.25	1723.65	4958.37	3374.17	572.66	131.95	45,152.25	278,353.70
2019 a-w	8362.46	4150.80	5317.32	4871.12	788.57	179.20	107,152.55	226,615.16
Zone 3. Tenosique								
2017 s-s	3080.31	190.15	3569.70	2561.97	762.54	19.49	51,663.70	126,387.18
2017 a-w	2245.22	365.40	4435.46	2900.94	622.71	86.30	87,333.25	90,146.37
2018 s-s	4568.87	156.32	3933.85	1603.03	423.14	10.12	77,259.58	101,880.44
2018 a-w	1856.00	290.86	3060.32	3871.29	593.40	159.43	76,436.89	101,967.70
2019 s-s	4675.46	142.43	2885.48	3174.07	625.46	52.29	34,752.37	141,926.51
2019 a-w	2269.37	452.24	3038.07	2497.06	708.77	64.25	87,729.06	93,745.83

**Table 8 sensors-22-04729-t008:** Land-use coverage classified by RF. Coverage in hectares.

Season	CC	SC	WB	LR	UA	SA	TR	Others
Zone 1. Emiliano Zapata								
2017 s-s	2769.57	1127.54	2931.90	472.56	316.34	10.78	11,841.73	39,760.49
2017 a-w	2097.25	400.71	5091.82	1015.73	359.71	279.25	21,870.46	28,115.99
2018 s-s	1369.75	577.45	3135.05	205.12	270.71	12.66	18,826.37	34,833.81
2018 a-w	1925.99	498.32	2978.68	610.61	346.99	350.18	23,395.11	29,125.03
2019 s-s	2307.29	1,330.62	1500.89	315.94	316.96	44.63	11,915.94	41,364.37
2019 a-w	2444.86	488.26	2853.61	789.13	400.51	124.95	21,574.19	30,555.41
Zone 2. Balancán								
2017 s-s	12598.43	5191.36	5752.22	5840.99	832.99	32.41	28,248.56	299,238.95
2017 a-w	6960.48	4725.05	8167.37	6800.37	524.80	284.70	119,955.66	210,317.24
2018 s-s	5077.67	2661.62	6598.98	893.01	333.37	55.21	110,099.69	232,016.35
2018 a-w	5848.83	4291.23	6134.84	6918.09	689.55	536.95	101,651.32	231,665.08
2019 s-s	8608.91	4436.02	957.58	2227.57	495.57	122.34	42,612.84	294,275.07
2019 a-w	9502.74	3217.59	5739.21	5667.76	692.72	308.39	109,065.48	235,420.01
Zone 3. Tenosique								
2017 s-s	4036.99	3230.39	3782.10	1596.70	495.61	10.09	47,177.75	127,905.39
2017 a-w	3203.48	1243.79	4774.91	4049.55	460.66	273.88	86,276.84	87,951.92
2018 s-s	2879.73	1664.45	4011.87	830.06	288.98	25.66	86,444.81	92,089.46
2018 a-w	3307.66	1555.56	3313.37	4636.41	517.43	317.57	77,226.23	97,360.76
2019 s-s	3651.14	3821.04	2948.24	1967.18	516.83	49.17	41,891.19	133,390.22
2019 a-w	4646.64	1454.98	3256.64	3336.57	577.33	123.29	88,130.18	86,709.40

**Table 9 sensors-22-04729-t009:** Land-use coverage classified by CART. Coverage in hectares.

Season	CC	SC	WB	LR	UA	SA	TR	Others
Zone 1. Emiliano Zapata								
2017 s-s	7842.29	4393.24	2267.24	314.17	326.06	121.26	13,546.98	30,419.69
2017 a-w	5441.09	2383.54	4353.35	1764.45	443.26	287.30	20,304.49	24,253.44
2018 s-s	8962.11	1710.50	2633.75	236.01	247.73	49.14	17,263.76	28,127.93
2018 a-w	4711.87	1614.52	1688.28	1194.54	378.51	1098.5	20,915.14	27,629.57
2019 s-s	5958.21	3524.67	1718.43	372.18	285.36	97.77	10,477.66	36,765.65
2019 a-w	5683.84	2510.76	1738.65	1306.61	459.65	913.41	19,569.59	27,048.42
Zone 2. Balancán								
2017 s-s	52,951.22	32,589.6	6820.26	2452.76	1890.40	1095.26	54,479.32	205,457.03
2017 a-w	24,318.22	22,800.81	7246.54	6072.87	835.25	612.87	116,901.50	178,947.87
2018 s-s	52,631.68	13,250.7	6634.00	606.79	381.43	149.91	120,140.88	164,940.43
2018 a-w	27,318.82	13,204.62	5092.27	7121.10	867.07	889.55	98,020.03	205,222.44
2019 s-s	39,514.52	26,606.6	5807.00	1546.87	471.27	258.93	5200.33	238,330.39
2019 a-w	29,281.72	15,707.49	3216.92	4647.01	1070.98	2476.39	110,215.11	191,120.28
Zone 3. Tenosique								
2017 s-s	21,547.47	13,259.8	3231.72	1199.76	744.97	252.75	59,488.63	88,510.64
2017 a-w	12,161.45	9266.72	4325.92	3491.13	584.33	284.85	83,586.10	74,534.53
2018 s-s	25,649.19	6545.0	4,153.16	572.72	273.31	107.84	91,325.50	59,608.26
2018 a-w	13,428.54	5993.69	2798.79	4093.97	546.92	277.96	75,955.14	85,140.02
2019 s-s	15,836.97	13,948.9	3379.60	1638.31	487.28	174.50	44,401.90	108,368.25
2019 a-w	13,933.62	8180.23	2161.82	2671.43	689.34	1112.83	83,780.06	75,705.71

**Table 10 sensors-22-04729-t010:** Percentage of corn and sorghum crop error by each classification method.

	Corn			Sorghum		
**Season**	**SVM**	**RF**	**CAR**	**SVM**	**RF**	**CAR**
	Zone 1. Emiliano Zapata					
2017 s-s	7.9%	62.26%	86.67%	6.32%	73.57%	93.21%
2017 a-w	6.11%	36.1%	75.37%	7.73%	19.39%	86.44%
2018 s-s	7.9%	25.75%	88.65%	12.42%	95.15%	98.36%
2018 a-w	2.87%	29.86%	70.92%	9.51%	80.93%	94.11%
2019 s-s	5.41%	49.07%	80.38%	10.4%	96.24%	98.58%
2019 a-w	5.41%	191.84%	74.92%	7.82%	76.44%	95.41%
	Zone 2. Balancán					
2017 s-s	2.95%	18.5%	80.61%	6.71%	51.11%	92.21%
2017 a-w	3.21%	8.85%	249.37%	5.54%	14.81%	82.34%
2018 s-s	1.7%	129.51%	77.42%	4.96%	95.97%	99.19%
2018 a-w	5.39%	35.51%	70.98%	4.85%	9.76%	70.67%
2019 s-s	3.45%	31.92%	72.25%	7.34%	63.79%	93.96%
2019 a-w	3.41%	14.9%	72.38%	4.61%	23.44%	74.71%
	Zone 3. Tenosique					
2017 s-s	7.7%	29.15%	86.72%	–	–	–
2017 a-w	7.66%	33.01%	82.35%	7.55%	72.74%	96.34%
2018 s-s	7.2%	47.99%	88.84%	–	–	–
2018 a-w	6.68%	47.45%	87.05%	10.68%	83.15%	95.62%
2019 s-s	8.6%	17.9%	72.81%	18.68%	96.85%	99.13%
2019 a-w	6.07%	13.37%	71.11%	6.87%	71.06%	94.85%

## Data Availability

Not applicable.

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
