# Peer review of "Mapping of Land Cover with Optical Images, Supervised Algorithms, and Google Earth Engine"

_sensors, 2022, doi:10.3390/s22134729_

Round 1

Reviewer 1 Report

The manuscript is well written, and data has a good presentation. However, the subject and methodology do not bring substantial scientific advances. Please, check below some considerations and suggestions to improve the paper.

Abstract – more specific information regarding the achieved results is required.

L26. This affirmation is vague. Please, add reference or delete.

The background information (sections 2.1 to 2.4) has a general and basic content for the journal audience. Some parts (e.g., Sentinel bands and vegetation indices used) should be included in the methodology, while others could be deleted, especially when the authors explain about Landsat, which will not be used in the analysis.

Section 2.5 – The related work section could be improved. There are hundreds of papers that used sentinel-2 image and performed land use mapping using vegetation indices. Explore them. At the end, it is important to mention what is the contribution of your paper compared to all papers mentioned in the section. What is the innovation of your paper?

Figure 2 – include map’s legend.

L217 – Which reduction method did you use?

L225-226 –How this data was created? Automatically? Visual interpretation? This database should be better explained.

Tables 6 and 7 – specify that the numbers in the tables represent hectares (?)

L274-275 – How is the ground truth? The same points mentioned before (30%)? Field data? This should be explicit in the methodology.

L292-299 – this is repeated information.

Discussion – This section is a bit superficial. Please improve it.

Author Response

Thank you very much for your comments and suggestions. This has strengthened the work. Attached document.

Reviewer 2 Report

This paper focuses on land cover mapping through supervised classification algorithms such as SVM, random forest, and CART. The paper is well-written, however it presents techniques already studied and very much used for land cover mapping, hence it does not provide any scientific novelty. I would suggest to review these few things:

(a) Improve the introduction adding more information as to why this paper is important.

(b) Improving the section of the previous work, which is mostly a review of remote sensing and satellites, but does not present a thorough presentation of the current techniques employed for land cover mapping.

Author Response

Thank you very much for your comments and suggestions. This has strengthened the work. Attached document

Reviewer 3 Report

First of all, the article is quite well written, but it should be consolidated in each part. Also, I am skeptical about the use of spectral indices and supervised algorithms used for classification.

Section 2.1

.L.55 the sensors provide 13 bands ... but 12 is presented in the table 1

Section 2.2

T.1 Please highlite wich bands are used, actually it's only presented at the end of the article L.311

Section 2.3

L.90 since few years, advances in spectral indices have been made. The use of spectrale indices if intereseting when dealing with unsupervised classification. It's not the case of this study.

It is well established that NDVI values are considerably influenced by number of factors like atmospheric conditions, scale of the imagery, vegetation moisture, soil moisture, overall vegetative cover, differences in soil type, variability in incident solar radiation, radiometric response characteristics of the sensor etc… see :

  1. Geerken, R., Batikha, N., Celis, D., and E. Depauw. 2005. Differentiation of rangeland vegetation and assessment of its status: field investigations and MODIS and SPOT VEGETATION data analyses. International Journal of Remote Sensing 26(20).
  2. Geerken, R., Zaitchik, B., and J.P. Evans. 2005. Classifying rangeland vegetation type and coverage from NDVI time series using Fourier Filtered Cycle Similarity. International Journal of Remote Sensing 26(24):5535-5554.
  3. Evans, J.P. and R. Geerken. 2006. Classifying rangeland vegetation types and coverage using a Fourier component based similarity measure. Remote Sensing of the Environment 105(1):1-8.
  4. Knight, J.F., R.L. Lunetta, J. Ediriwickrema, and S. Khorram, 2006. Regional Scale Land-Cover Characterization using MODIS-NDVI 250 m Multi-Temporal Imagery: A Phenology Based Approach. GIScience and Remote Sensing, 43(1), 1-23.

For example, NDVI itself is poorly defined, like most remote sensing indices. It is defined completely empirically and uses only a small amount of discriminating information from the spectra. These indices can be replaced by any machine learning algorithm. A simple linear regression would give better results than any remote sensing index. So use ML technics instead of these indices. like Genetic algorithm and Deep learning. see :

  1. Synthesis of Vegetation Indices Using Genetic Programming for Soil Erosion Estimation  
  2. DeepIndices: Remote Sensing Indices Based on Approximation of Functions through Deep-Learning, Application to Uncalibrated Vegetation Images  

Standard spectral indices are a transformation of input images, which take a limited amount of information from the input. Thus spectral indices lead to a loss of discriminating information essential for classification during the transformation. Spectral indices are strongly limited by this simple fact. So, actually, the assertion of the L.90 is wrong and the choice of spectral indices with supervised learning is questionable.

E.3 The equation 3 do not respect the same definition as E.1 E.2 ...
Please use the same variable Red or R. Blue or B, ... etc

E.5 The NDWI use the SWIR band, but the table 1 have SWIR1 and SWIR2 which one is used ?

L.119-122 Only 3 classification algorithm are presented. But as mentionned earlier, genetic algorithm and deep learning are also possible.

Section 2.5 

This section should be consolidated.

L.148-154 The first paragraph present the article [50] it's correct, but does the presented results of 80% OA are computed only from spectral indices ? does it use other spectral bands ?

L.155-161 The second paragraph presente the article [51], which classification algorithm are used ? what performances are optained ?

L.162-167 Which performances ? Does the results computed only from spectral indices ?

L.168-171 same remarks for article [53]

L.171-175 which spectral bands are used ? what performances ?

Section 3.1

F.1 The first figure prsent a spectral band selection, which do not appear in the paper. The percentiles calculation is difficult to understand, which data are in the input, etc. We will see that later. What is the "corrected image collection classification" (with a misspelling). Finnaly, in the paper, the dataset is splitted into training (70%) and test (30%). What is this Validation step ?

Section 3.2

L.199-206 This paragraph present a small algorithm to select images with the minimum of cloud. But it is never evaluated ! Since spectral indices are used in this article, why a cloud indices was not used ?

  • ISPRS Journal of Photogrammetry and Remote Sensing
    Volume 144, October 2018, Pages 235-253 : Cloud/shadow detection based on spectral indices for multi/hyperspectral optical remote sensing imagery

Section 3.3

L.215-222 In fact, this paragraph is not understandable. A figure should be proposed to help the reader. For example, the output contains the same number of spectral bands as the input collection. Okay, so 12 or 13 ... ? You probably add the spectral indices, so ~18. And you add 5 percentiles ... but from what ? What is the number of outputs of each percentile ? I don't understand why there are 78 bands at the end.

Section 3.4

L.240-244 The choice of separated sping-summer and autum-winter is interesting so you use classical classification algorithm. But please note that a time-serie classification algorithm could have been used is this study.

T.3 Please note that the dataset is starting to be highly unbalanced. This implie that the used metric (OA) should be analysed carefully.

L.248 please note the training and test set

Section 4

L.262 please note that the test set is called "validation" with is an error

T.10 Tell me how you can have a percentage of error greater than 100%. ? (191.84%, 129.51%, 249.37%). The mean of error for the SVM is of 5.55%, this could have been mentionned for all model.

T.9 As mentioned above, SVMs have an average error percentage of about 5.55%. How can you have an overall accuracy of 99.73%? 0.9966 is a typo? or maybe it's ~1% AO ... in which case, it doesn't work ... The KI show almost a perfect classification which is in contraction with the result presented in T.10.

Section 5

L.309-313 Why all spectral bands and spectrals indices are not used ? Why it's presented only at the and of the article and not in the methodology section ?

Section 6

L.336 Overall accuracy : 0.99% 0.95% 0.92% ... close to 1%, does it work?

Author Response

(The authors gave the same response as above.)

Round 2

Reviewer 1 Report

The modifications applied improved the quality of the manuscript. However, it still requires English review. 

Author Response

Thank you very much for your observations. This has improved our work.
The entire document has been analyzed to improve English
